# The cost-effectiveness of scaling-up rapid point-of-care testing for early infant diagnosis of HIV in southern Zambia

Gatien De Broucker[1], Phillip P. Salvatore[2], Simon Mutembo[3], Nkumbula Moyo[4], Jane N. Mutanga[5], Philip E. Thuma[4], William J. Moss[1,2], Catherine G. Sutcliffe[1,2]*

1 Department of International Health, Johns Hopkins Bloomberg School of Public Health, Baltimore, MD, United States of America, 2 Department of Epidemiology, Johns Hopkins Bloomberg School of Public Health, Baltimore, MD, United States of America, 3 Provincial Medical Office, Choma, Zambia, 4 Macha Research Trust, Choma, Zambia, 5 Livingstone Central Hospital, Livingstone, Zambia

* csutcli1@jhu.edu

## Abstract

### Introduction

Early infant diagnosis (EID) and treatment can prevent much of the HIV-related morbidity and mortality experienced by children but is challenging to implement in sub-Saharan Africa. Point-of-care (PoC) testing would decentralize testing and increase access to rapid diagnosis. The objective of this study was to determine the cost-effectiveness of PoC testing in Southern Province, Zambia.

### Methods

A decision tree model was developed to compare health outcomes and costs between the standard of care (SoC) and PoC testing using GeneXpert and m-PIMA platforms. The primary health outcome was antiretroviral treatment (ART) initiation within 60 days of sample collection. Additional outcomes included ART initiation by 12 months of age and death prior to ART initiation. Costs included both capital and recurrent costs. Health outcomes and costs were combined to create incremental cost effectiveness ratios (ICERs).

### Results

The proportion of children initiating ART within 60 days increased from 27.8% with SoC to 79.8–82.8% with PoC testing depending on the algorithm and platform. The proportion of children initiating ART by 12 months of age increased from 50.9% with SoC to 84.0–86.5% with PoC testing. The proportion of HIV-infected children dying prior to ART initiation decreased from 18.1% with SoC to 3.8–4.6% with PoC testing. Total program costs were similar for the SoC and GeneXpert but higher for m-PIMA. ICERs for PoC testing were favorable, ranging from $23–1,609 for ART initiation within 60 days, $37–2,491 for ART initiation by 12 months of age, and $90–6,188 for deaths prior to ART initiation. Factors impacting the costs of PoC testing, including the lifespan of the testing instruments and integrated utilization of PoC platforms, had the biggest impact on the ICERs. Integrating utilization across

**Data Availability Statement:** All relevant data are within the paper and its Supporting Information files.

**Funding:** This work was supported by a grant from the National Institutes of Allergy and Infectious Disease (https://www.niaid.nih.gov/)(WJM, grant number R01AI116324. The funder had no role in the study design, data collection and analysis, decision to publish, or preparation in the manuscript.

**Competing interests:** The authors have declared that no competing interests exist.

programs decreased costs for the EID program, such that PoC testing was cost-saving in some situations.

## Conclusion

PoC testing has the potential to improve linkage to care and ART initiation for HIV-infected infants and should be considered for implementation within EID programs to achieve equity in access to HIV services and reduce HIV-related pediatric morbidity and mortality.

## Introduction

In 2019, an estimated 1.8 million children were living with HIV globally [1], with approximately 90% residing in sub-Saharan Africa. While great progress was made over the last decade in reducing the number of children acquiring HIV by scaling up prevention of mother-to-child transmission (PMTCT) programs, 150,000 children were newly infected in 2019 [1]. Preventing the morbidity and mortality experienced by infants living with HIV requires timely diagnosis and treatment. However, only 60% of infants born to women living with HIV in 2019 were tested by two months of age [1].

In sub-Saharan Africa, early infant diagnosis (EID) of HIV infection is challenging as it requires nucleic acid-based testing, which is typically only available in laboratories in urban areas. Consequently, EID commonly occurs through a centralized testing process, involving transport of blood samples from the clinics to the central laboratory and test results back to the clinics. While this process is recommended to take no more than 4 weeks [2], there are many steps and opportunities for delays, such that longer turnaround times of up to three months have been reported, with some infants never receiving their results [3–7].

Over the last decade, there has been progress in developing point-of-care (PoC) EID tests to decentralize the testing process. These tests are intended to be performed at health facilities where HIV-exposed infants are brought for testing and require a lower level of training and resources than laboratory-based testing. Consequently, PoC tests offer the potential for same-day testing and results and rapid treatment initiation. Two randomized controlled trials conducted in Zimbabwe and Malawi to evaluate PoC testing found that almost all (98–99%) caregivers in the PoC arm received same-day results compared to none in the standard of care (SoC) arm, and 89–91% of HIV-infected infants initiated treatment within 60 days of sample collection in the PoC arm compared to only 13–42% in the SoC arm [8, 9]. An additional multi-country observational study found similar results [10]. These studies support the utility of PoC platforms for rapidly diagnosing HIV and linking HIV-infected infants to care.

Transitioning from laboratory to PoC testing for EID will require significant investment in new technologies and training. Several studies have found PoC testing to be cost-effective compared to the SoC [10, 11]. These studies provide some evidence to support investment in PoC testing but do not provide details about the costs of a PoC EID program or consider different implementation strategies on which to base policy decisions. This study was conducted to provide information on the health benefits and costs of implementing a PoC EID program with currently available PoC platforms and with different testing algorithms and implementation strategies in Southern Province, Zambia.

## Materials and methods

### Overview and setting

The health benefits and costs of implementing PoC testing for EID were modeled for Southern Province, Zambia. Zambia had an estimated HIV prevalence of 12% in 2016 [12] and 66,000 children living with HIV in 2019, including 6,000 newly-infected children [1]. At the time of the study, guidelines recommended virologic testing at birth, 6 weeks, and 6 months of age, with additional serologic testing at 9, 12, 18, and 24 months of age and ≥6 weeks after breast-feeding cessation [13].

A decision tree model was developed for Southern Province, where studies were conducted to estimate health and cost parameters. The model assumed an annual cohort of 7,500 infants requiring EID (see S1 Methods for further details) and was run over a 5-year time horizon (the expected lifespan of the PoC platforms) for a total of 37,500 infants. The model simulated testing of infants at birth, 6 weeks and 6 months of age, the ages requiring nucleic acid-based testing and thus impacted by PoC testing. Infants could enter at and drop out after any of the three testing points. HIV-exposed infants could acquire HIV until the 6 month test, with a risk dependent on their age and the ART and PMTCT status of the mother and child. Infants who acquired HIV but were not diagnosed or did not initiate ART within 60 days of sample collection experienced age-dependent HIV-related mortality. HIV-infected infants not receiving ART by 6 months of age were followed until 12 months of age for ART initiation and HIV-related mortality.

Two PoC platforms, four testing algorithms, and three implementation models were compared to centralized laboratory-based testing, the current SoC in Zambia.

### Testing platforms and algorithms

In the SoC, nucleic acid-based testing at the central laboratory was modeled using the COBAS® AmpliPrep/COBAS® TaqMan® system (Roche Diagnostics, Risch-Rotkreuz, Switzerland). To illustrate an idealized comparison between PoC platforms and a diagnostic gold standard, a simplifying assumption was made that the SoC platform had a sensitivity and specificity of 100%.

Two PoC platforms were modeled: the m-PIMA (Abbott Laboratories, Lake Forest, Illinois) and the Xpert HIV-1 Qual performed on the GeneXpert IV (Cepheid Inc, Sunnyale, California) [14–16]. The platforms are intended to be used by trained laboratory technicians or healthcare workers with results available in 56 (m-PIMA) or 95 (GeneXpert) minutes [14, 15]. In addition to EID, both platforms are prequalified for HIV viral load testing [17, 18], and the Xpert platform can additionally be used for tuberculosis testing.

The testing algorithms considered were based on current testing guidelines in Zambia and included immediate ART initiation and confirmatory testing after a first positive result [13]. In the SoC (S1 Methods in Fig 1), blood samples were collected at the clinics and transported to the centralized laboratory for testing with results returned some period of time later. If positive, ART was initiated immediately and a second blood sample was collected for confirmatory testing at the centralized laboratory. Four algorithms were modeled for PoC testing (S1 Methods in Figs 2–5). All algorithms started with an initial PoC test performed at the clinic and then differed on confirmatory testing for initial positive tests. The first and second algorithms (labelled PoC3 and PoC2+SoC) assumed that confirmatory testing would be performed at the clinic with a second PoC test, with a tie-breaker test performed at the central laboratory (PoC2 +SoC) or at the clinic (PoC3). In both algorithms, ART initiation was assumed to occur after results of the tie-breaker test were available. The third algorithm assumed that confirmatory

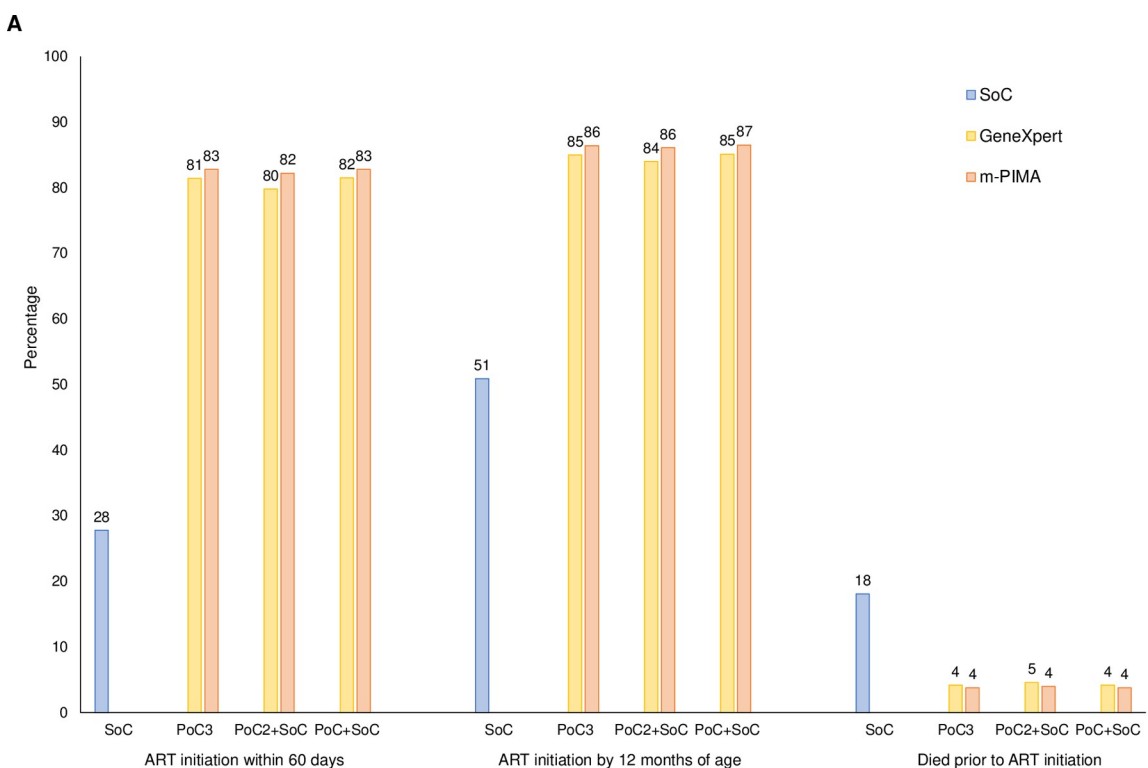

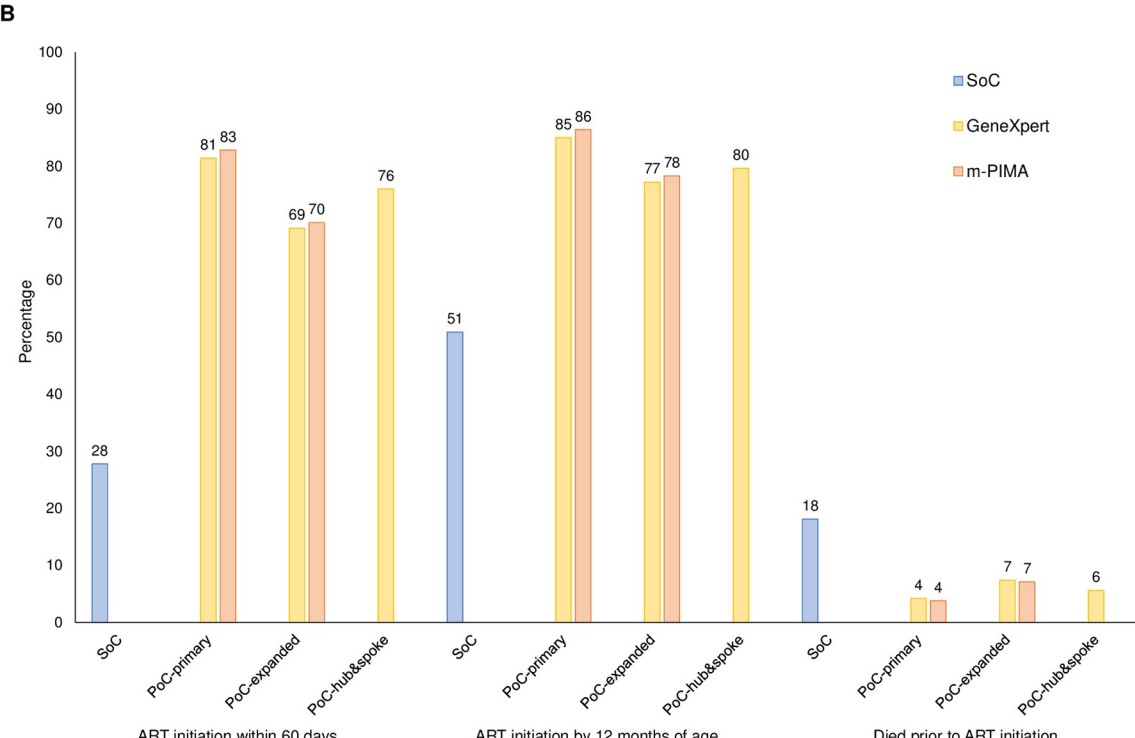

**Fig 1.** Health outcomes for standard of care and point-of-care testing by (A) testing algorithm and (B) implementation model. ART: antiretroviral therapy; PoC: point-of-care testing; SoC: standard of care testing. Note: (A) testing algorithms are presented for the primary implementation model; (B) implementation models are presented for the PoC3 algorithm.

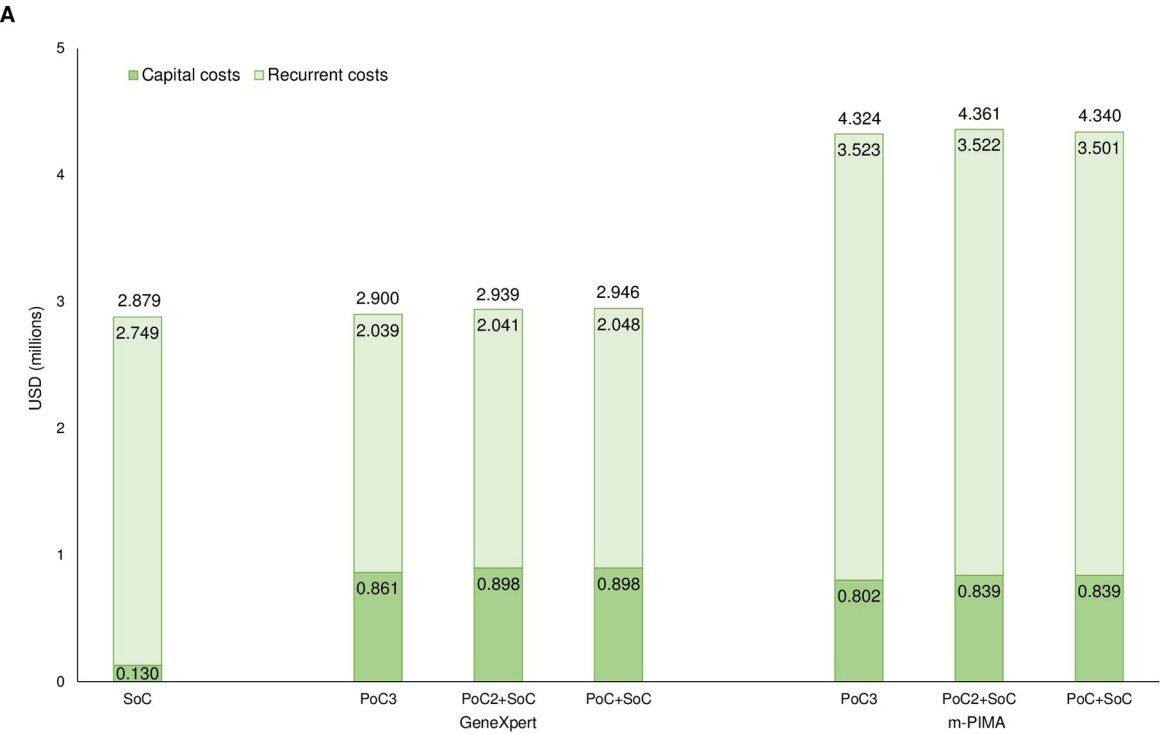

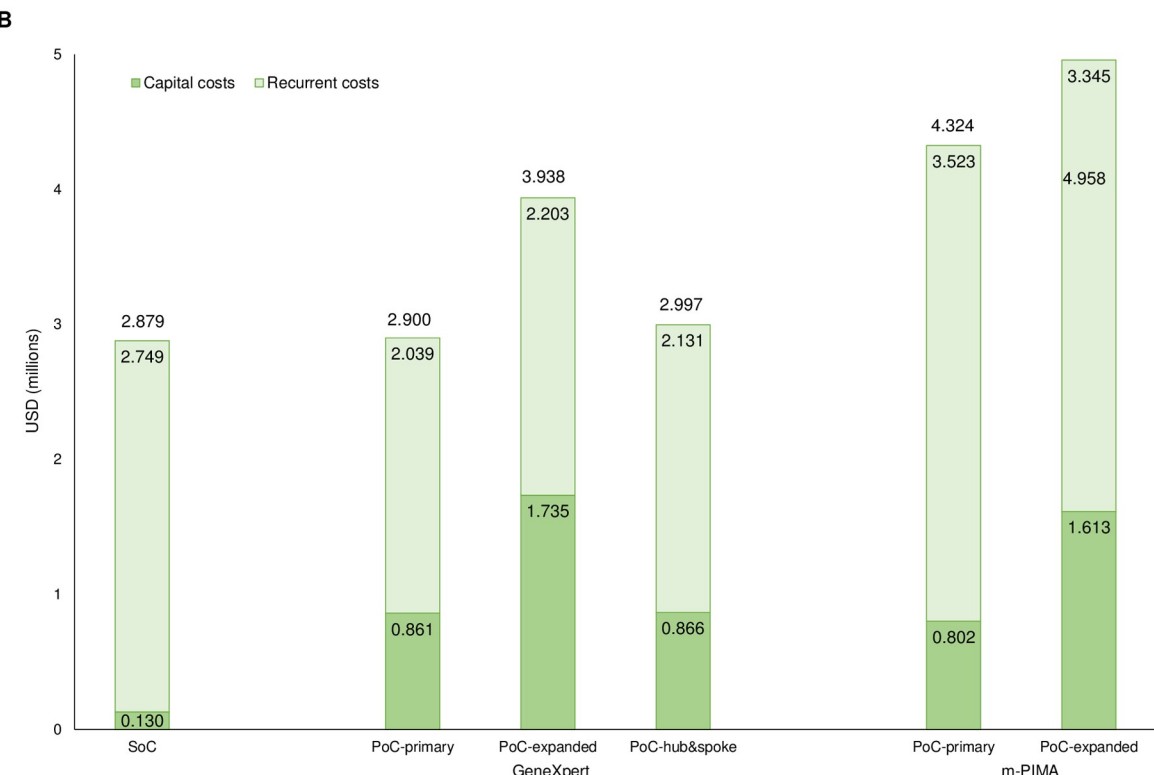

**Fig 2.** Capital and recurrent costs for standard of care and point-of-care testing by (A) testing algorithm and (B) implementation model. PoC: point-of-care testing; SoC: standard of care testing. Note: (A) testing algorithms are presented for the primary implementation model; (B) implementation models are presented for the PoC3 algorithm.

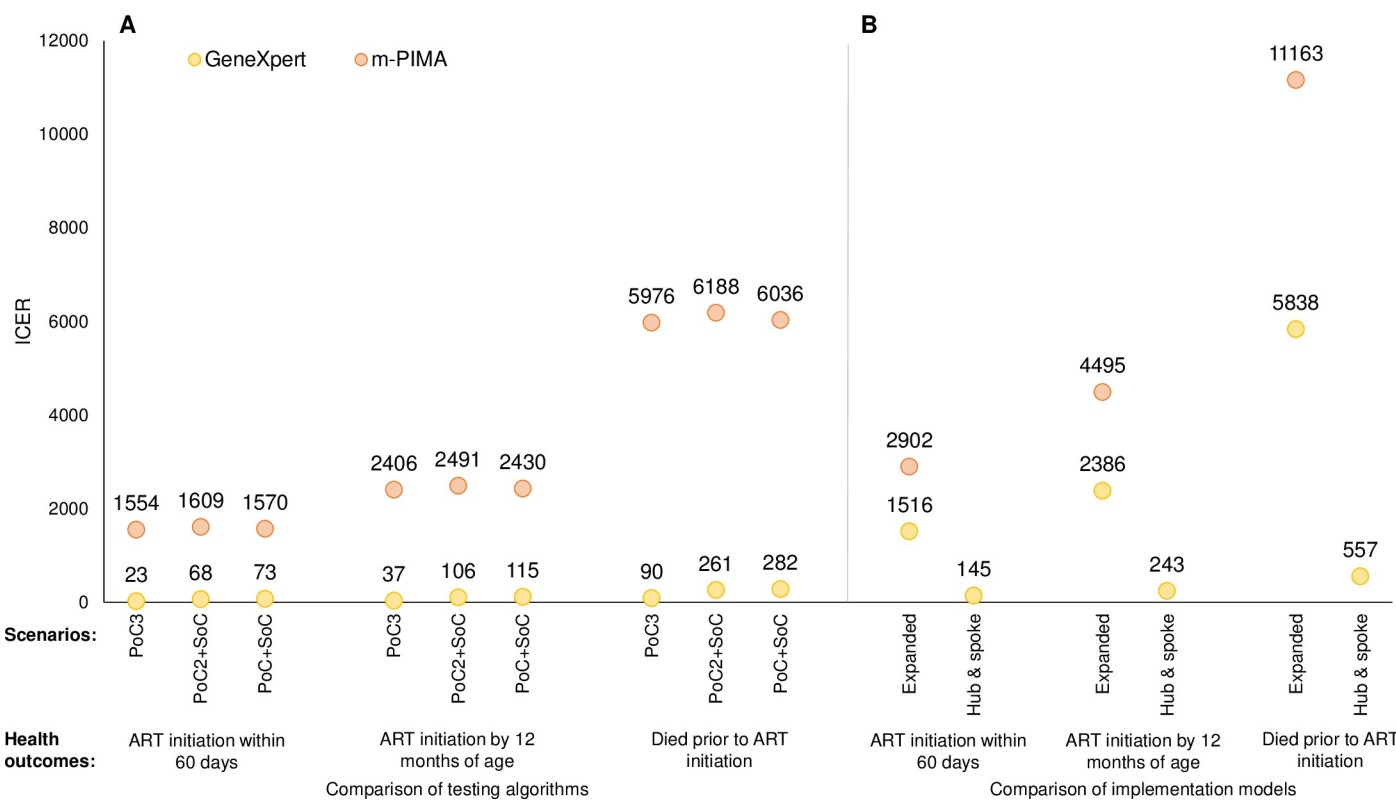

**Fig 3.** Incremental cost-effectiveness ratios for point-of-care testing by (A) algorithm and (B) implementation model. ART: antiretroviral therapy; ICER: incremental cost-effectiveness ratio; PoC: point-of-care testing; SoC: standard of care testing. Note: (A) testing algorithms are presented for the primary implementation model; (B) implementation models are presented for the PoC3 algorithm.

testing would be performed at the central laboratory (PoC+SoC). ART initiation was assumed to occur after the initial positive test result, with treatment cessation in the event of a negative confirmatory test. The last algorithm did not include confirmatory testing and was based on a single PoC test, which may occur in practice [8]. As confirmatory testing is recommended, results of this algorithm are only presented in the (S2 Table).

## Implementation models

The primary implementation model included placement of PoC platforms at selected facilities (n = 40) in the province based on their volume of samples collected for EID testing (minimum average of 1.5 samples per week; see S1 Methods for further details). These facilities were assumed to cover 61% of the HIV-exposed infant population; all other HIV-exposed infants were assumed to be referred to these facilities for testing.

The second implementation model included expanded access to PoC testing with PoC platforms placed at a larger number of facilities (n = 74; minimum average of 3.5 samples per month) covering a larger proportion of the HIV-exposed infant population (77%). The remaining 23% was assumed to be tested under the SoC.

The third implementation model included a hub-and-spoke approach, with the 40 facilities from the primary model serving as PoC testing hubs. For the 61% of HIV-exposed infants served by the hubs, testing was considered point-of-care, with results available on the same day. For the remaining 39% of the population, testing was considered near point-of-care, with DBS cards transported to the hubs and test results transported to the facilities with the goal of

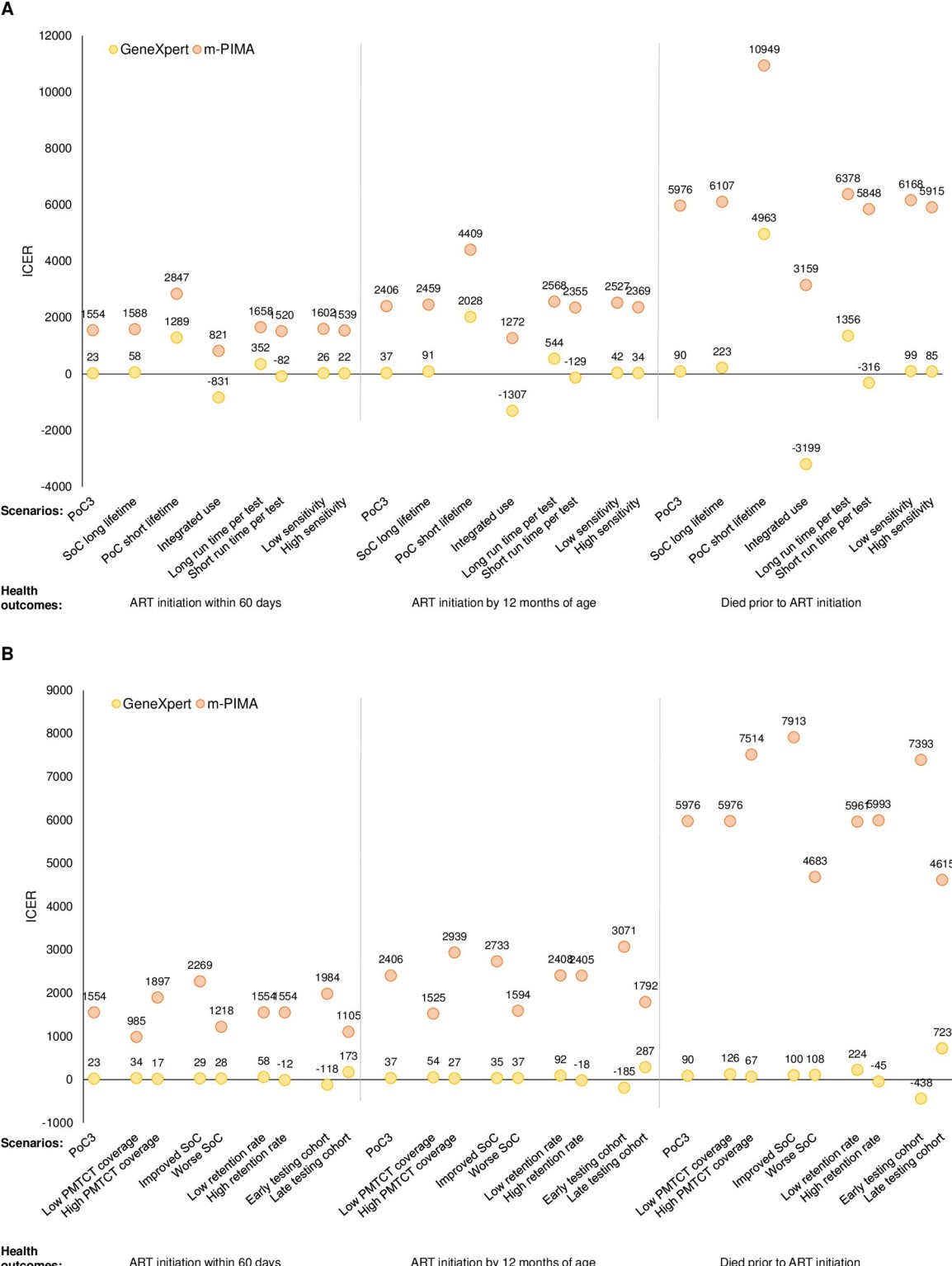

**Fig 4.** Incremental cost-effectiveness ratios for sensitivity analyses of (A) intrinsic and (B) external factors affecting point-of-care testing. ART: antiretroviral therapy; ICER: incremental cost-effectiveness ratio; PoC: point-of-care testing; SoC: standard of care testing. Note: All sensitivity analyses were performed with the PoC3 algorithm and primary implementation model. For (A) and (B), PoC3 represents the results from the primary analysis with the baseline model parameters. For (A) PoC short lifespan refers to the time period covered by the manufacturer's warranty; integrated use refers to use of the PoC machines across programs (HIV viral load and tuberculosis testing);

long and short run time per test refers to the amount of staff time spent for sample collection (including pre-test counseling) and running the test. For (B) improved SoC refers to the proportion of children initiating ART within 60 days after SoC (ICER shown for PoC3 algorithm for the primary implementation model compared to SoC); retention rate refers to the proportion of children returning for subsequent testing at a later age. See S1 Table for the list of parameter values.

returning to mothers within four-weeks of sample collection. As the m-PIMA is not currently approved for use with DBS cards, this model was only considered for GeneXpert.

## Model parameters and data collection

Epidemiologic parameters for the model were estimated from the literature and data collected from two studies on EID conducted in Southern Province, Zambia (Tables 1 and S1): the Early Infant Diagnosis (EID) study (conducted from 2013–2015) [19, 20] and the Novel Screening for Exposed Babies (NSEBA) study (conducted from 2016–2019) [7, 21]. For further details, see the (S1 Methods).

As part of the NSEBA study, the costs of sample collection and testing for both PoC and the SoC were also collected. Using an ingredient-based costing approach, the operating costs for each instrument were estimated in their specific settings (PCR in a central laboratory, GeneXpert and m-PIMA in clinics). Cost data were obtained from Livingstone Central Hospital for lab-based testing in 2016 and from vendor invoices and administrative data for GeneXpert and m-PIMA in 2018 and 2019. The lifespan of the instruments, estimated at 5 years, was based on details provided by the manufacturer. For the SoC, additional cost estimates for

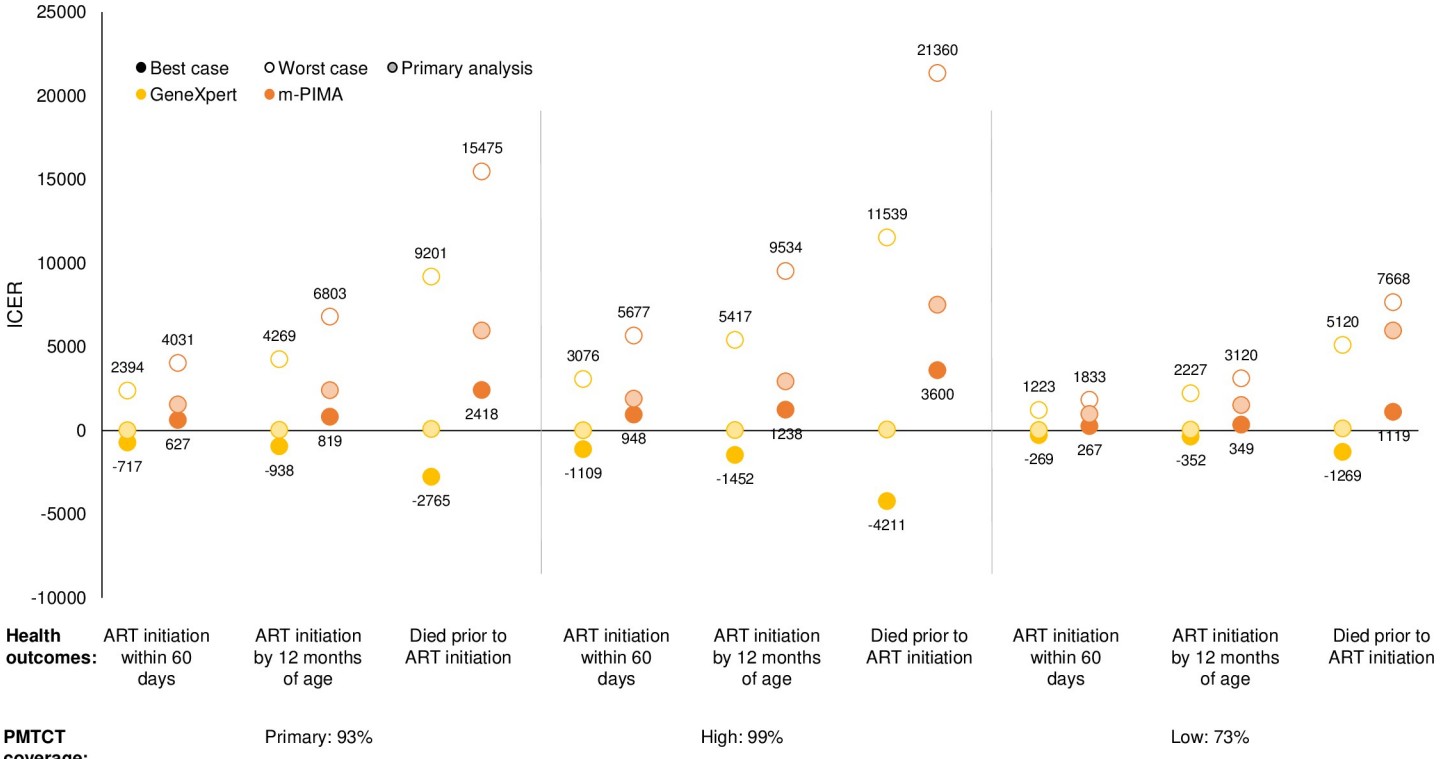

**Fig 5. Incremental cost-effectiveness ratios for multivariate sensitivity analysis showing best and worst cases by PMTCT coverage.** ART: antiretroviral therapy; ICER: incremental cost-effectiveness ratio; PMTCT: prevention of mother-to-child transmission. Note: All sensitivity analyses were performed with the PoC3 algorithm and primary implementation model. See S1 and S7 Tables for the list of parameter values for the primary analysis, best case, and worst case scenarios.

**Table 1. Summary of select model parameters (for a complete list of model parameters see S1 Table).**

| Parameter | Primary analysis value | Sensitivity analysis range | Source |
|---|---|---|---|
| **Annual number of children tested** | 7,500 | | MoH program data |
| **Proportion of children entering the cohort** | | | |
| At birth | 0.40 | 0.15–0.80 | NSEBA Study |
| At 6 weeks of age | 0.45 | 0.45–0.10 | NSEBA Study |
| At 6 months of age | 0.15 | 0.40–0.10 | NSEBA Study |
| **PMTCT coverage** | 0.93 | 0.73–0.99 | [22] |
| **Risk of mother-to-child transmission of HIV** (range dependent on age at testing) | | | |
| Mother received PMTCT | 0.01–0.08 | | EID Study [23–29] |
| Mother did not receive PMTCT | 0.02–0.3 | | EID Study [27, 29–33] |
| **Probability of ART initiation** | | | |
| After SoC, on treatment within 60 days / ever | 0.30 / 0.55 | 0.13–0.43 / 0.35–0.65 | NSEBA Study [7–9, 19] |
| After PoC testing, on treatment within 60 days / ever | 0.90 / 0.94 | | [8, 9] |
| **Probability of returning for subsequent EID testing** | | | |
| Mother received PMTCT | 0.8 | 0.75–0.85 | NSEBA Study |
| Mother did not receive PMTCT | 0.6 | 0.35–0.85 | NSEBA Study |
| For tie-breaker test a week later after discrepant first and second tests (PoC3 algorithm only) | 0.97 | | Assumption |
| **Risk of HIV-related mortality** (range dependent on age) | 0.012–0.33 | | [34] |
| **SoC costs and characteristics** | | | |
| Sensitivity / specificity | 1 / 1 | | Assumption |
| Capital costs | $200,895 | | NSEBA Study, CHAI |
| Lifetime of instrument | 5 years | 3 years | NSEBA Study, CHAI |
| Utilization rate of instrument | 0.15 | 0.10–1.00 | Assumption |
| Cost of reagents | $15.43 | | NSEBA Study |
| Cost of blood collection supplies | $1.94 | | NSEBA Study |
| Wastage rate[a] | 1% | | NSEBA Study |
| Probability of caregiver receiving results | 0.9 | | NSEBA Study |
| **GeneXpert costs and characteristics** | | | |
| Sensitivity / specificity | 0.968 / 0.9991 | 0.9268–0.9895 / not varied | [16] |
| Capital costs | $30,130 | | NSEBA Study |
| Lifetime of instrument | 7 years | 3 years | NSEBA Study, CHAI |
| Utilization rate of instrument | 1.0 | 0.10–1.00 | Assumption |
| Cost of reagents | $14.90 | | NSEBA Study |
| Cost of blood collection supplies | $0.46 | | NSEBA Study |
| Time spent to run each test | 0.25 hours | 0.17–0.5 | NSEBA Study |
| Wastage rate[a] | 9% | | NSEBA Study CHAI |
| Probability of caregiver receiving results | 1.0 | | Assumption |
| **m-PIMA costs and characteristics** | | | |
| Sensitivity / specificity | 0.99 / 0.9997 | 0.9645–0.9988 / not varied | [16] |
| Capital costs | $20,042 | | NSEBA Study |
| Lifetime of instrument | 5 years | 2 years | NSEBA Study, CHAI |
| Utilization rate of instrument | 1.0 | 0.10–1.00 | Assumption |
| Cost of reagents | $31.52 | | NSEBA Study |
| Cost of blood collection supplies | $1.35 | | NSEBA Study |

*(Continued)*

**Table 1.** (Continued)

| Parameter | Primary analysis value | Sensitivity analysis range | Source |
|---|---|---|---|
| Time spent to run each test | 0.25 hours | 0.17–0.5 | NSEBA Study |
| Wastage rate[a] | 9% | | CHAI [35, 36] |
| Probability of caregiver receiving results | 1.0 | | Assumption |
| **Cost of staff time for blood collection for SoC and PoC** | | | |
| Cost per hour of registered nurse | $4.00 | | NSEBA Study |
| Time spent per test on sample collection (includes pre-test counseling) | 0.6 hours | 0.5–1.0 | NSEBA Study |

[a] Proportion of tests that yield an error or invalid result and need to be re-run.

ART: antiretroviral therapy; EID: early infant diagnosis; MoH: Ministry of Health; PMTCT: prevention of mother-to-child transmission; PoC: point-of-care; SoC: standard of care.

freight and installation of the instruments were based on estimates provided by the Clinton Health Access Initiative (CHAI).

## Health outcomes

Health outcomes were considered up to treatment initiation, death, or 12 months of age. For each algorithm, implementation model, and PoC platform, the proportions of HIV-infected children treated within 60 days of sample collection, treated by 12 months of age, and dying prior to ART initiation were considered as health outcomes. In addition, the proportion of children treated with ART but falsely diagnosed with HIV was also included.

## Costs

Costs were reported in 2018 USD. Costs collected in 2016 for the NSEBA study were corrected assuming an average annual inflation of 8% (16.64% 2-years cumulative) [37]. We used a conversion rate of 1 USD = 10 ZMW [38]. For each platform, total costs, including both one-time capital costs and recurrent costs per test, were calculated (see S1 Methods for detailed information on estimated costs). Capital costs included costs of the instrument, maintenance, freight, insurance, inspection, handling, and clearance, shipping, and distribution. Recurrent costs included reagents and supplies for testing, supplies for specimen collection, transportation of samples, salaries for specimen collection, processing, and testing, and waste management.

For the SoC, the PCR instrument was assumed to be used for both EID and HIV viral load testing. The capital costs were adjusted based on the utilization rate of the instrument for EID testing to reflect the share of these costs contributed by the EID program. For the PoC platforms, use of the instrument was modeled in the primary analysis assuming that it would be used solely for EID testing (100% utilization rate).

## Primary and sensitivity analyses

For all analyses, health outcomes and costs were reported for each PoC platform. In addition, health outcomes and costs were combined to produce incremental cost-effectiveness ratios (ICER), comparing PoC testing to the SoC. The primary analysis included a comparison of health outcomes, costs, and ICERs across testing algorithms for the primary implementation model and across implementation models for the PoC3 algorithm. All calculations and analyses were performed using R software version 3.5.0 (R Core Team, Vienna, Austria).

Sensitivity analyses were conducted to evaluate the influence of model parameters on the health outcomes, costs, and ICERs. Using the primary implementation model and the PoC3 algorithm, a univariate sensitivity analysis was performed by iteratively re-running the model, each time with a single parameter set to a more extreme value than in the primary analysis (Table 1 and S1 Table). Parameters of interest included the lifespan of the SoC and PoC platforms, sensitivity of the PoC platforms, the time spent by clinic staff on PoC testing, the utilization rate of the PoC platforms, PMTCT coverage, the likelihood of ART initiation with SoC, distribution of infants entering the testing schedule at different ages, and retention of infants in the testing schedule. For the lifespan of the PoC instrument, the shorter lifespan was assumed to be the number of years under warranty by the manufacturer (GeneXpert: 3 years; m-PIMA: 2 years). For the utilization rate of the PoC platforms, integrated use of the PoC platforms across disease programs was assumed. In this setting the PoC instruments would be used for HIV viral load testing (both platforms) and tuberculosis testing (GeneXpert only) and the capital costs were adjusted according to the utilization rates for EID. Prioritization of the instruments for EID was assumed, with no impact of integrated use on health outcomes for EID. A multivariate sensitivity analysis was then performed for each PMTCT coverage setting (primary, low, high) to obtain a worst (all parameters of interest set to their worst value) and best (all parameters of interest set to their best value) case scenario for PoC compared to SoC.

## Results

### Comparison of testing algorithms in the primary implementation model

For a population of 37,500 children tested over 5 years (7,500 children tested annually), the model yielded 1,692 HIV-infected children, of whom 92.6% were diagnosed with SoC and 90.5–92.0% were diagnosed with PoC testing. Use of PoC testing significantly improved health outcomes for HIV-infected children (Fig 1A and S2 Table). The proportion of children initiating ART within 60 days increased from 27.8% with SoC to 79.8–82.8% with PoC testing, depending on the algorithm and platform. The proportion of children initiating ART by 12 months of age increased from 50.9% with SoC to 84.0–86.5% with PoC testing. The proportion of HIV-infected children dying prior to ART initiation decreased from 18.1% with SoC to 3.8–4.6% with PoC testing. The proportion of children receiving ART who were falsely diagnosed was low with PoC testing (compared to none with SoC), ranging from 0% to 0.4% (S2 Table). Both PoC platforms and all algorithms produced similar health outcomes.

Recurrent costs accounted for most of the total program costs and differed by platform: $38.07 per test for the SoC, $27.91 per test for GeneXpert, and $48.28 per test for m-PIMA. Total program costs were similar for the SoC and GeneXpert but higher for m-PIMA (Fig 2A).

Given improved health outcomes and similar or higher costs, ICERs for PoC testing were favorable, ranging from $23 to $1,609 for ART initiation within 60 days, $37 to $2,491 for ART initiation by 12 months of age, and $90 to $6,188 for deaths prior to ART initiation (Fig 3A). Given similar health outcomes but higher program costs, ICERs for m-PIMA were higher than for GeneXpert. While ICERs across testing algorithms were similar, the PoC3 algorithm performed best.

In sensitivity analyses, factors impacting the costs of PoC testing, including the lifespan of the testing instruments, integrated utilization of the PoC platforms, and the time spent per test, had the biggest impact on the ICERs (Fig 4A and 4B and S3, S4 and S5 Tables). A shorter lifespan doubled the capital costs and resulted in higher ICERs. Integrating use of the PoC platforms across programs decreased the share of the capital costs contributed by the EID program and resulted in lower ICERs, such that PoC testing was cost-saving in some situations. Decreasing the time required for sample collection and testing decreased recurrent costs and

also resulted in lower ICERs, such that PoC testing was cost-saving (Fig 4A and S3 Table). Best and worst case scenarios for each PMTCT setting are presented in Fig 5 and S7 Table.

## Comparison of implementation models

The primary implementation model was the most cost-effective programmatic option, dominant over both the expanded access and hub-and-spoke models (Fig 3B; see S6 Table). In the model with expanded access to PoC testing, health outcomes were improved compared to the SoC but were lower than the primary model due to shared provision of testing between PoC platforms and the SoC (Fig 1B). Increasing the number of facilities providing PoC testing and adding some testing through the SoC approximately doubled the capital costs, such that total program costs were higher than the primary model (Fig 2B). Consequently, ICERs for this model were 2–66 times higher than the primary model (Fig 3B).

In the hub-and-spoke model, health outcomes were improved compared to the SoC but were lower than the primary model due to delays in diagnosis and loss to follow-up at the spoke facilities (Fig 1B). Program costs were only marginally higher than for the primary model due to the added costs of transporting DBS samples and results between the hub and spoke facilities (Fig 2B). Consequently, ICERs were 2–6 times higher than the primary model (Fig 3B).

## Discussion

This study evaluated health outcomes and costs associated with PoC testing for EID of HIV infection. Across all testing algorithms and implementation models considered, PoC testing resulted in significant health benefits, with more rapid identification and treatment initiation and decreased HIV-related mortality for HIV-infected infants. With PoC platforms used exclusively for EID, PoC testing with both platforms had higher costs than the SoC. Integrating use of the PoC platforms across disease programs decreased the capital costs for the EID program such that PoC testing was cost-saving with GeneXpert.

The results of this analysis are consistent with other studies that also found PoC testing to improve EID and linkage to care and to be cost-effective. In a multi-country observational study, the proportion of infants with HIV initiating ART within 60 days of sample collection increased from 43.3% with conventional testing to 92.3% with PoC testing, and the cost per test result returned within 30 days was lower for PoC ($27.24) than for conventional testing ($131.02) [10]. In a study from Zimbabwe, PoC testing increased life-expectancy for HIV-infected infants and the ICER for EID comparing PoC assays to conventional testing was $680 per year of life saved, which was considered to be cost-effective by the authors [11]. While all studies used different methodologies and outcomes making direct comparison challenging, the health benefits and cost-effectiveness of PoC testing were consistent.

This study adds to the literature on the utility of PoC testing by evaluating practical implementation questions such as how confirmatory testing is performed, either through additional PoC testing or involvement of central labs, and how to effectively test HIV-exposed infants with varying access to testing sites. First, this study considered several testing algorithms and found that varied levels of involvement of the central labs in confirmatory testing produced similar health outcomes and costs. Second, this study considered three implementation models that made different assumptions about how to provide PoC testing for HIV-exposed infants who can access testing sites and how those infants who cannot would be tested. Each model considered placement of the PoC platforms at facilities based on their volume of HIV-exposed infants. However, the primary and hub-and-spoke models provided PoC platforms for facilities directly covering only 60% of HIV-exposed infants and therefore had lower costs and were

more cost-effective than the expanded model, which directly covered 77% of HIV-exposed infants. The primary and hub-and-spoke models made different assumptions about how to cover the remaining 40% of the population, either through referrals or transport of DBS cards, respectively, with little impact on health outcomes and costs. All testing algorithms and models involving PoC testing resulted in improved health outcomes for HIV-infected infants but with different costs of testing and requirements for testing infrastructure.

Finally, this study also evaluated both intrinsic and external factors influencing the cost-effectiveness of PoC testing. The most influential factors included the lifespan of the instrument, staff time for sample collection and testing, and integrated use of the PoC platforms across programs. The lifespan of the instrument depends on the environment in which it is used, including infrastructure, maintenance, and personnel. Deficiencies in any of these areas will result in a shorter lifespan of the instrument and significantly higher capital costs for the program, which will decrease the cost-effectiveness of PoC testing. The staff time required for sample collection and testing will be in part determined by the efficiency of clinic and lab staff and the wastage rate for the PoC platform. With additional training and experience, the required staff time may be decreased to minimize the human resource costs associated with testing.

Integrated use of the PoC instruments with other programs resulted in a significant reduction in the capital costs to be paid by the EID program, which significantly increased the cost-effectiveness of PoC testing for EID such that it was even cost-saving in certain situations. One study evaluating the use of the GeneXpert platform for multi-disease testing found it to be feasible, with the ability to increase access to EID, HIV viral load, and tuberculosis testing in Zimbabwe [39]. In settings with a high burden of tuberculosis and HIV, such as Zambia (tuberculosis incidence of 333 per 100,000 in 2019 [40] and HIV prevalence of 12% in 2016 [12]), PoC platforms that can be shared across programs to support diagnostic testing and treatment monitoring for other conditions would be beneficial. While integrated use has the potential to decrease costs, there is also the potential for a negative impact on health outcomes if use of the platform for other tests delays EID testing. This could be mitigated by prioritizing EID testing over other tests, as was assumed in this analysis. As tests for additional conditions are developed and implemented with these platforms, PoC testing for EID will only become more cost-effective.

The results of the analysis should be interpreted in the context of the model parameters and assumptions that were made. As demonstrated in the sensitivity analyses, both the health outcomes and costs were influenced by the selected values of the parameters. Therefore, these results may not be applicable to settings with significantly different characteristics or where the costs for SoC or PoC testing are significantly different. In addition, the model evaluated the costs and benefits of PoC testing for linking HIV-infected infants to care and treatment to guide policy decisions around testing. The model did not consider longer term treatment outcomes and costs for those infants linked to care. As stated earlier, the study from Zimbabwe considered the impact of PoC testing on the entire lifespan of HIV-infected infants and found that PoC testing increased life-expectancy and still was cost-effective compared to conventional testing [11].

As the feasibility and cost-effectiveness of implementing PoC testing for EID depends on many factors, decisions about incorporating PoC testing into local diagnostic networks, including the optimal platform, algorithm and model to be used, will need to account for local needs and capacity. While the decision to initiate PoC testing may be made at a national or regional level, how best to implement testing may vary more locally. In addition, as SoC testing will likely remain a component of the EID diagnostic network, as either the primary testing method where PoC testing is not appropriate or feasible or as the confirmatory testing method

depending on the algorithm selected, continued efforts will be needed to improve SoC testing and address the existing challenges for early diagnosis and linkage to care.

## Conclusions

In summary, PoC testing has the potential to significantly improve EID for all HIV-exposed infants, leading to linkage to care and early ART initiation for HIV-infected infants. While implementing PoC testing for EID will require an investment in new technologies and infrastructure, integrating use of PoC platforms across disease programs will increase their cost-effectiveness. EID programs should consider implementing PoC testing to achieve equity in access to HIV services and reduce HIV-related pediatric morbidity and mortality.

## Supporting information

**S1 File.**
(ZIP)

**S1 Methods. Supplemental methods.** This document describes in detail the methods used for the study and assumptions made in the analysis.
(DOCX)

**S1 Table. Model inputs.**
(DOCX)

**S2 Table. Health outcomes and costs by testing platform and algorithm for the primary implementation model.**
(DOCX)

**S3 Table. Sensitivity analysis of intrinsic factors influencing costs.**
(DOCX)

**S4 Table. Sensitivity analysis of health outcomes and costs by PMTCT coverage.**
(DOCX)

**S5 Table. Sensitivity analysis of external factors influencing health outcomes and costs.**
(DOCX)

**S6 Table. Health outcomes and costs by implementation model.**
(DOCX)

**S7 Table Multivariable sensitivity analysis: A) parameter values and B) health outcomes and costs.**
(DOCX)

## Acknowledgments

We thank the Clinton Health Access Initiative for providing platform deployment and cost data to inform our analysis.

We thank the NSEBA study teams for their assistance and the children and their parents for participating in the study. We also thank the District and Provincial Health Directors in Choma and Livingstone, and the clinic staff at the study sites for supporting the conduct of the study.

## Author Contributions

**Conceptualization:** William J. Moss, Catherine G. Sutcliffe.

**Data curation:** Simon Mutembo.

**Formal analysis:** Gatien De Broucker, Phillip P. Salvatore, Catherine G. Sutcliffe.

**Funding acquisition:** William J. Moss.

**Methodology:** Gatien De Broucker, Phillip P. Salvatore.

**Project administration:** Nkumbula Moyo, Jane N. Mutanga, Catherine G. Sutcliffe.

**Supervision:** Philip E. Thuma, William J. Moss.

**Writing – original draft:** Gatien De Broucker, Catherine G. Sutcliffe.

**Writing – review & editing:** Phillip P. Salvatore, Simon Mutembo, Nkumbula Moyo, Jane N. Mutanga, Philip E. Thuma, William J. Moss.

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
