## [Decision Letter · Decision Letter 0]

6 Jan 2021

PONE-D-20-38117

The cost-effectiveness of scaling-up rapid point-of-care testing for early infant diagnosis of HIV in southern Zambia

PLOS ONE

Dear Dr. Sutcliffe,

Thank you for submitting your manuscript to PLOS ONE. After careful consideration, we feel that it has merit but does not fully meet PLOS ONE’s publication criteria as it currently stands. Therefore, we invite you to submit a revised version of the manuscript that addresses the points raised during the review process.

We look forward to receiving your revised manuscript.

Kind regards,

Claudia Marotta

Academic Editor

PLOS ONE

Journal Requirements:

Additional Editor Comments:

dear authors follow reviewer suggestion to improve your paper

Reviewers' comments:

Reviewer's Responses to Questions

**Comments to the Author**

1. Is the manuscript technically sound, and do the data support the conclusions?

Reviewer #1: Partly

Reviewer #2: Yes

2. Has the statistical analysis been performed appropriately and rigorously? 

Reviewer #1: I Don't Know

Reviewer #2: Yes

3. Have the authors made all data underlying the findings in their manuscript fully available?

Reviewer #1: Yes

Reviewer #2: Yes

4. Is the manuscript presented in an intelligible fashion and written in standard English?

Reviewer #1: Yes

Reviewer #2: Yes

5. Review Comments to the Author

Reviewer #1: Summary :

The purpose of this study was to determine whether scaling up of rapid PoC testing for EID in Southern Province , Zambia was cost effective.

Current SoC for EID in Zambia was testing in laboratories in urban areas and samples had to be transported from clinics to the centralized labs for testing.

This resulted in delay in EID, resulting in delay in ART initiation and increased

morbidity and mortality of infants because of this delay.

In this study, the process of studying cost effectiveness involved using two PoC platforms (m-PIMA and GeneXpert IV), Four algorithms (PoC 3-meaning all three tests for EID were performed at PoC site, PoC2+SoC where two tests were performed at PoC and the third confirmatory test at SoC, PoC+SoC where one test was done at PoC and one test at SoC, and the fourth algorithm where only one test was performed at PoC without a confirmatory test.) three implementation models (first one was having 40 PoC sites covering 61 % of the HIV exposed infants, the second was expanded to 70 PoC sites covering 77% of the population, with remaining going to the SoC sites and the third was a hub and spoke approach with 40 PoC sites also receiving DBS cards from other clinics and reporting results within four weeks(only GeneXpert platform). The health outcomes for each of the PoC platforms, algorithms and implementation model were ART initiation by 60 days of sample collection, treatment by 12 months of age and mortality of infant prior to ART initiation. In addition, health outcomes and costs were combined to produce incremental cost effectiveness ratios (ICER) comparing PoC testing to SoC. The primary analysis included a comparison for health outcomes, costs and ICERS for all four algorithms the primary implementation model and also for all three implementation models using only the PoC3 algorithm.

The results showed that with both PoC platforms across all four algorithms using the primary implementation model there was a significant increase in infants initiating ART within 60 days, within 12 months and a significant decrease in infant mortality prior to staring ART compared to SoC. Further the incidence of false positivity with PoC was less than with SoC.

With regard to health costs both capital and recurrent ,the costs were comparable for PoC (only GeneXpert) and SoC but higher for m-PIMA.The ICERS for PoC were favourable and the factors that impacted the ICER were life span of the equipment and its intergrated use across programs .

In conclusion, PoC testing across all algorithms, using the primary implementation model showed significant improvement in health outcomes with decrease in mortality. However health costs remain an issue- The SoC and PoC by GeneXpert were comparable but high for PoC by m-PIMA. By integrating the PoC platforms across disease programs, cost effectiveness could be achieved.

1. The intent of the study is good- whether PoC testing is cost effective and has better health outcomes than current SoC.

2.This is a detailed study and many variables are being studied- two PoC platforms, four algorithms, three implementation models, ICERS, internal and external factors that influence ICERs, worst and best case scenarios for PMTCT coverage at different settings of high and low. There is a lot of data that is being presented and I dont have the expertise to comment on the appropriateness of the data. That is why I have written partly as my response to the first question in the review. I would suggest that a statistician review the data.

3. The authors have made all the data available either in the manuscript or in the supplemental data.

4. The authors have not defined the time and period of study.

5. I could not find reference no 38 in the manuscript.

6. The discussion could include a line or two about incidence of Tuberculosis in southern province ,Zambia

to account for use of PoC platform geneXpert across diseases .

7. Line 149 to 152 - The selection criteria should be included in the body of the manuscript rather than in the supplementary data.

Reviewer #2: Review of the article titled: The cost-effectiveness of scaling-up rapid point-of-care testing for early infant diagnosis of HIV in southern Zambia

Though tremendous progress has been made in the diagnosis of HIV among infants, challenges remain in certain communities; centered mostly around demand creation for testing, access to testing platforms and quick return of test results for clinical action. World Health Organozation’s prequalification for the use of both GeneXpert and mPIMA platforms to support near point of care and point of care testing respectively for infant diagnosis has been seen a game changer, supported with earlier data from Mozambique, Zimbabwe and Malawi. Within this period, there has been strong recommendations to move diagnostics from silo to multiplexing, particularly within the integrated diagnostic network optimization to tackle multiple diseases in a space already occupied by centralized or standard of care (SoC) platforms. The choice to either use point of care or SoC or both in a complementary fashion has sometimes been challenged by many factors to include cost-effectiveness of operations. The current study that evaluated health outcomes and costs associated with PoC testing for HIV diagnosis among infants is timely and highly welcomed. The fact that the authors were able to show that PoC testing with both platforms had higher costs than the SoC and integrated use of the PoC platforms across disease programs decreased the capital costs for the EID program such that PoC testing was cost-saving with GeneXpert adds to other existing knowledge in this field and provides further knowledge on how to address issues around confirmatory testing which so far has been very low and resulting in HIV misdiagnosis among these infants.

The design and outcomes of this study is adequate to meet the stated objectives, congratulations to the authors, however, they should consider including in the abstract and discussion sections the fact that point of care testing should be considered and seen to occur within an integrated diagnostic optimized network and complimentarily with SoC in infants diagnosis. It is not in every setting that point of care platforms must be used for infant diagnose; the decision to use point of care should be informed by the outcomes of an optimized network, addressing all systemic issues and prioritizing public health impact of quick diagnosis and access to care of these infants. In effect, where SoC is the most appropriate platform for diagnosis, for example when large sample volumes are to be tested, systemic challenges to include long turnaround time for the return of results should be addressed rather that substituting for point of care. This study itself confirms the fact that SoC is not more expensive when compared with point of care.

6. PLOS authors have the option to publish the peer review history of their article (what does this mean?). If published, this will include your full peer review and any attached files.

Reviewer #1: No

Reviewer #2: **Yes: **George Alemnji

---

## [Author Response · Author response to Decision Letter 0]

5 Feb 2021

Reviewer #1: 

4. The authors have not defined the time and period of study.

The model was run over a five-year time horizon (as outlined on lines 106-107) and is anchored in 2018 when the majority of the costs were collected (as outlined on lines 179-181 and 192). 

Data were collected in two studies (NSEBA study and EID study) with details provided primarily in the Supplementary Methods. To assist the reader, the period of performance for each study was added to the main text of the manuscript (lines 167-169). 

5. I could not find reference no 38 in the manuscript.

We thank the reviewer for identifying this omission. We have corrected the issue and reference 38 is now included on line 194.

6. The discussion could include a line or two about incidence of Tuberculosis in Southern Province, Zambia to account for use of PoC platform geneXpert across diseases.

The discussion was revised to include a sentence about the utility of multi-disease platforms in high burden setting, with a reference to the burden of tuberculosis and HIV in Zambia (lines 369-372).

7. Line 149 to 152 - The selection criteria should be included in the body of the manuscript rather than in the supplementary data.

The selection criteria for the primary and expanded access implementation models were added to the main text of the manuscript (lines 149-150 and 154-155).

Reviewer #2: 

The design and outcomes of this study is adequate to meet the stated objectives, congratulations to the authors, however, they should consider including in the abstract and discussion sections the fact that point of care testing should be considered and seen to occur within an integrated diagnostic optimized network and complimentarily with SoC in infants diagnosis. It is not in every setting that point of care platforms must be used for infant diagnose; the decision to use point of care should be informed by the outcomes of an optimized network, addressing all systemic issues and prioritizing public health impact of quick diagnosis and access to care of these infants. In effect, where SoC is the most appropriate platform for diagnosis, for example when large sample volumes are to be tested, systemic challenges to include long turnaround time for the return of results should be addressed rather that substituting for point of care. This study itself confirms the fact that SoC is not more expensive when compared with point of care.

We agree with the Reviewer and thank them for raising this point. PoC testing is not feasible or desirable in all settings, and where implemented it will need to be incorporated into an overall program that still includes SoC testing. The cost-effectiveness of PoC testing depends on many factors, including how well the SoC performs. This was demonstrated in our sensitivity analysis, particularly for the m-PIMA platform. SoC testing may be the best option for certain settings or may continue to be part of the testing algorithm where PoC testing is implemented. Either way, implementation of PoC testing should not be viewed as a decision to stop working to improve laboratory testing. A functional and efficient laboratory system will have benefits for EID as well as many other conditions. The Discussion section was revised to include these points (lines 389 to 397).

---

## [Decision Letter · Decision Letter 1]

23 Feb 2021

The cost-effectiveness of scaling-up rapid point-of-care testing for early infant diagnosis of HIV in southern Zambia

PONE-D-20-38117R1

Dear Dr.Sutcliffe,

We’re pleased to inform you that your manuscript has been judged scientifically suitable for publication and will be formally accepted for publication once it meets all outstanding technical requirements.

Kind regards,

Claudia Marotta

Academic Editor

PLOS ONE

Additional Editor Comments (optional):

dear authors congratulations

Reviewers' comments:

Reviewer's Responses to Questions

**Comments to the Author**

1. If the authors have adequately addressed your comments raised in a previous round of review and you feel that this manuscript is now acceptable for publication, you may indicate that here to bypass the “Comments to the Author” section, enter your conflict of interest statement in the “Confidential to Editor” section, and submit your "Accept" recommendation.

Reviewer #1: All comments have been addressed

Reviewer #2: All comments have been addressed

2. Is the manuscript technically sound, and do the data support the conclusions?

Reviewer #1: Yes

Reviewer #2: Yes

3. Has the statistical analysis been performed appropriately and rigorously? 

Reviewer #1: N/A

Reviewer #2: Yes

4. Have the authors made all data underlying the findings in their manuscript fully available?

Reviewer #1: Yes

Reviewer #2: Yes

5. Is the manuscript presented in an intelligible fashion and written in standard English?

Reviewer #1: Yes

Reviewer #2: Yes

6. Review Comments to the Author

Reviewer #1: This is my second review of the article. All my recommendations have been met. This paper may be accepted for publication.

Reviewer #2: I am fully satisfied with the revised version of the manuscript. I congratulate the authors for their efforts in publishing this data which is critical to support diagnostics in resource limited settings.

7. PLOS authors have the option to publish the peer review history of their article (what does this mean?). If published, this will include your full peer review and any attached files.

Reviewer #1: No

Reviewer #2: No

---

## [Editor Report · Acceptance letter]

26 Feb 2021

PONE-D-20-38117R1 

The cost-effectiveness of scaling-up rapid point-of-care testing for early infant diagnosis of HIV in southern Zambia 

Dear Dr. Sutcliffe:

I'm pleased to inform you that your manuscript has been deemed suitable for publication in PLOS ONE. Congratulations! Your manuscript is now with our production department. 

Kind regards, 

on behalf of

Dr. Claudia Marotta 

Academic Editor

PLOS ONE